# A reversible phospho-switch mediated by ULK1 regulates the activity of autophagy protease ATG4B

N. Pengo[1], A. Agrotis[1], K. Prak[1], J. Jones [1] & R. Ketteler [1]

Upon induction of autophagy, the ubiquitin-like protein LC3 is conjugated to phosphatidy-lethanolamine (PE) on the inner and outer membrane of autophagosomes to allow cargo selection and autophagosome formation. LC3 undergoes two processing steps, the proteolytic cleavage of pro-LC3 and the de-lipidation of LC3-PE from autophagosomes, both executed by the same cysteine protease ATG4. How ATG4 activity is regulated to co-ordinate these events is currently unknown. Here we find that ULK1, a protein kinase activated at the autophagosome formation site, phosphorylates human ATG4B on serine 316. Phosphorylation at this residue results in inhibition of its catalytic activity in vitro and in vivo. On the other hand, phosphatase PP2A-PP2R3B can remove this inhibitory phosphorylation. We propose that the opposing activities of ULK1-mediated phosphorylation and PP2A-mediated dephosphorylation provide a phospho-switch that regulates the cellular activity of ATG4B to control LC3 processing.

[1] MRC Laboratory for Molecular Cell Biology, University College London, Gower Street, London WC1E 6BT, UK. Correspondence and requests for materials should be addressed to R.K. (email: r.ketteler@ucl.ac.uk)

Autophagy is a cellular process that engulfs damaged organelles and cytoplasmic material in double membrane vesicles, which later fuse with lysosomes for degradation and recycling of their content. Most of the *ATG* (autophagy-related) genes involved in autophagy were identified in *Saccharomyces cerevisiae* over a decade ago, and many of their roles in autophagy, particularly at the early stages of autophagosome formation, have been clarified[1]. Among these is the yeast ubiquitin-like protein Atg8 or LC3B, one of its mammalian homologues. Upon induction of autophagy, a C-terminal glycine of LC3B is covalently conjugated to phosphatidyl-ethanolamine (PE) by the concerted action of ATG7, ATG3 and ATG12–5-ATG16L[2]. Membrane-anchored LC3-PE has multiple functions in autophagy. First, in *S. cerevisiae*, Atg8 is required for autophagosome formation and its abundance regulates autophagosome size[3]. Moreover, the identification of cargo receptors that bind through specialized domains to both LC3 and ubiquitin indicate a role for LC3 in selective autophagy[4]. Finally, binding of LC3 to FICOY suggests a role in autophagosome microtubule transport[5] while binding to VPS33 links it to autophagosome–lysosome fusion[6]. In addition to its regulation by conjugation, the Atg8/LC3 family of proteins are synthesized as an inactive proform with one or more amino acids shielding the C-terminal glycine and impeding lipidation[7]. A cysteine protease, Atg4, is thus needed to cleave these residues and allow exposure of the glycine and conjugation with PE[8]. Interestingly, the same protease is required at a later step to de-conjugate Atg8-PE from the outer membrane of autophagosomes probably preceding or just after lysosome fusion, as suggested by genetic evidence in yeast[9] and by immunoelectron microscopy and fluorescent microscopy in mammalian cells[10, 11].

A key regulator of autophagosome formation is Atg1/ULK1, a protein kinase that is activated early upon autophagy stimulation and is needed for the recruitment of other Atg proteins to the autophagosome formation site[12]. Interestingly, its kinase activity is dispensable for the recruitment of Atg proteins but is instead required for autophagosome formation[13, 14]. The finding that ULK1 is able to bind to the downstream autophagy protein LC3, via an LC3-interacting region, also suggests a role for Atg1/ULK1 in autophagosome formation[15]. Recently, the consensus sequences of surrounding residues that are phosphorylated by Atg1 have been identified[16]. These findings have been confirmed in mammals where a similar consensus sequence was described and additional ULK1 phosphorylation targets in the core machinery were identified[17].

In a proteomic study to delineate the autophagy protein network, it has been noted that ATG4B may be able to interact directly with ULK1, together with ATG101[18]. Here we investigated whether ULK1 was able to modulate ATG4B activity in mammalian cells. We show that ATG4B activity is controlled by reversible phosphorylation mediated by ULK1 and dephosphorylation by PP2A.

## Results

**ULK1 inhibits ATG4B activity and LC3B processing.** First, we investigated the effect of ULK1 expression on ATG4B activity. Using a luciferase-based assay that monitors ATG4B-mediated processing of LC3B2 by releasing luciferase into cellular supernatants,[19, 20] we found that overexpression of catalytic active ULK1 reduced the amount of released luciferase and thus of ATG4B activity when compared with a catalytically inactive ULK1 mutant (K46I)[21] when ATG4B was co-expressed in cells (Fig. 1a). To investigate whether this was a direct effect on ATG4B activity, we incubated recombinant active ULK1 with ATG4B and tested the hydrolase activity in vitro towards

an LC3B-GST (glutathione *S*-transferase) fusion construct. Compared with samples incubated with wild-type (WT) ATG4B, where free LC3 quickly accumulated indicating cleavage of the construct, in samples where ATG4B had been preincubated with ULK1 a lower level of free LC3 accumulated, indicating decreased ATG4B activity (Fig. 1b). In order to investigate whether the catalytic activity of ULK1 is required for this inhibition in ATG4B activity, we performed the LC3B-GST cleavage assay in the presence or absence of ATP. We show that cleavage of LC3B-GST is significantly lower in the presence of ULK1 and ATP compared with ULK1 without ATP (Figs. 1c, d). Thus ULK1 is able to inhibit LC3 processing by a direct effect on ATG4B, possibly by phosphorylation of a serine residue of ATG4B.

**ULK1 phosphorylates serine 316 of ATG4B.** In order to identify the target residue for ULK1-mediated phosphorylation of ATG4B, we investigated the residues that have been found to be phosphorylated on ATG4B in proteomic databases (www.phosphosite.org). Strikingly, ATG4B serine 316 has been shown to be phosphorylated in HeLa cells in vivo[22] and the sequence surrounding this residue matches the recently described ULK1 consensus motif[16, 17] with an enrichment in hydrophobic amino acids surrounding Ser316, conserved leucine/methionine residues at position −3 and an absence of charged amino acids at positions +1 to +3, which are instead hydrophobic (Fig. 2a). Importantly, ATG4B serine 316 is conserved in all four ATG4 isoforms and across eukaryotic species (Fig. 2a). To determine whether serine 316 is the target of ULK1 phosphorylation, we mutated the residue to alanine and tested the ability of ULK1 to phosphorylate ATG4B using in vitro phosphorylation assays. In the presence of radio-labelled $^{32}$P-ATP, we observed an increase in $^{32}$P incorporation in recombinant ATG4B upon incubation with purified ULK1, indicating that ULK1 is able to directly phosphorylate ATG4B. We found the amount of radioactivity incorporated in ATG4B S316A mutants was strongly reduced compared with WT ATG4B while, as a control, no difference was observed in the amount of ULK1 auto-phosphorylation (Fig. 2b). These results demonstrate that serine 316 is a target of ULK1 phosphorylation on ATG4B in vitro. It should be noted that an increase in ATG4B S316A $^{32}$P incorporation was nonetheless detected over time, suggesting that additional sites may be phosphorylated in response to ULK1, although this effect could also be an artifact from the in vitro phosphorylation assay. Therefore, in order to confirm that S316 is the main target for ULK1 phosphorylation, we produced a custom phospho-specific polyclonal antibody against ATG4B serine 316, anti-pATG4B (Ser316). Confirming our previous findings, we found that anti-pATG4B(Ser316) strongly reacted with purified ATG4B WT after incubation with ULK1 but not with the ATG4B S316A mutant, attesting to the specificity of the antibody (Fig. 2c). Interestingly, the antibody did not react with human ATG4A in the presence of ULK1 (Supplementary Fig. 1). However, this does not rule out that other ATG4 members are not phosphorylated by ULK1 as the phospho-antibody may not crossreact with the slightly different motif. In fact, another study has shown that yeast ATG4, which is strikingly different in sequence surrounding S316 (Fig. 2a) is indeed phosphorylated by ULK1 but at a different residue in proximity[23]. Next, in order to confirm whether ULK1-mediated phosphorylation of ATG4B on Ser316 occurs in cells, we co-transfected Halo-ATG4B with myc-ULK1 in HEK293T cells. Compared with samples transfected with a kinase inactive version of ULK1, the samples transfected with WT ULK1 displayed a strong increase in ATG4B Ser316 phosphorylation (Fig. 2d). This signal was reduced in samples expressing ATG4B S316A (Fig. 2d), as well as in protein lysates treated with

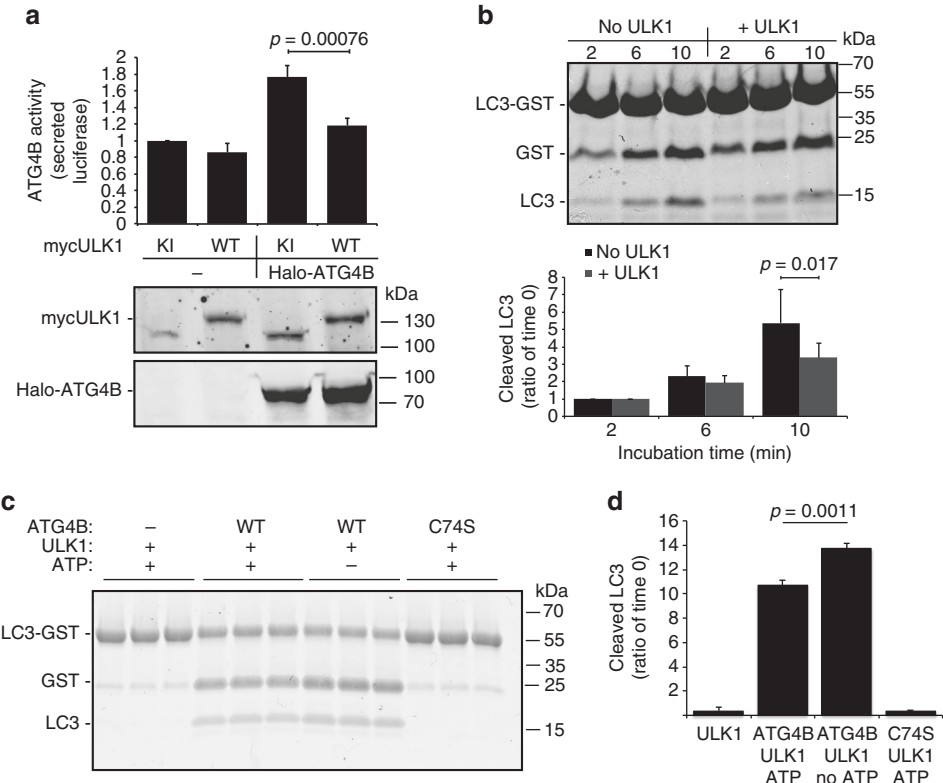

**Fig. 1** ULK1 inhibits ATG4B-mediated LC3 cleavage. **a** Average ATG4B activity in Actin-LC3-DelN Luciferase HEK293T obtained by measuring the secreted luciferase activity 48 h after transfection with the indicated constructs ($n = 3$, average ± s.d.) and representative immunoblot from one experiment showing expression of the different constructs. **b** GST-LC3 assay to measure in vitro activity of recombinant ATG4B after incubation with active recombinant ULK1. Representative gel and quantification of free cleaved LC3 ($n = 3$, average ± s.d.). **c** GST-LC3 assay to measure in vitro activity of recombinant ATG4B after incubation with active recombinant ULK1 in the presence or absence of 1 mM ATP. Representative gel showing three independent replicates for each condition and **d** quantification of free cleaved LC3 ($n = 3$, average ± s.d.)

recombinant phosphatase (Supplementary Fig. 2a), thus confirming that ULK1 is able to phosphorylate ATG4B on serine 316 in vivo. Finally, to confirm that ATG4B is phosphorylated on Ser316 also without ULK1 overexpression, we immunoprecipitated tagged ATG4B from WT or ULK1/2 double knock-out (DKO) mouse embryonic fibroblasts (MEFs) treated with okadaic acid to enhance the levels of Ser316-phosphorylated ATG4B (see explanation below). Indeed ATG4B from ULK1/2 DKO showed a 65% reduction reduction in phosphorylation on Ser316 (Fig. 2e), showing that endogenous ULK kinase is responsible for phosphorylation of ATG4B on Ser316 in cells. A residual, phosphorylation-dependent pATG4B(Ser316) antibody signal was still observed in ULK1/2 DKO MEFs (Supplementary Fig. 2b), suggesting that other kinases may be able to phosphorylate ATG4B at Ser316, albeit to a limited extent, or the antibody might mildly crossreact with other phosphorylation sites on ATG4B. Of interest, pATG4B(Ser316) antibody showed less overall reactivity in lysates not overexpressing ULK1 (Supplementary Fig. 2a) or in lysates from DKO MEF when compared with their WT counterpart (Fig. 2e, Input), suggesting a possible broader reactivity of the pATG4B(Ser316) antibody against ULK1 targets in line with the uniqueness of the ULK1 consensus phosphorylation motif[16].

**ATG4B Ser316 phosphorylation reduces LC3 binding and inhibits its catalytic activity.** The region surrounding serine 316 is conserved among species beyond the ULK1 phosphorylation motif, suggesting an important role for this region in ATG4B structure or function. Indeed, the co-crystal structure of ATG4B

and LC3[24] shows that serine 316 lies at the interface between ATG4B and LC3 and may be involved in hydrogen bonding between ATG4B and LC3B (Fig. 3a). We hypothesized that phosphorylation at this site might disrupt ATG4B activity by affecting the ATG4B–LC3 complex formation, thus explaining the observed loss of ATG4B catalytic activity upon ULK1 overexpression. To test the effects of ULK1-mediated ATG4B phosphorylation towards its catalytic activity, we mutated Ser316 to aspartate to mimic the charge given by the addition of a phosphate. First, we performed co-immunoprecipitation (IP) experiments of LC3 with ATG4B and its mutants S316A and S316D. We observed that both S316A and S316D showed reduced pull-down with endogenous LC3, indicating a reduced affinity for binding (Fig. 3b, *lower panel*). There was also reduced binding with overexpressed green fluorescent protein (GFP)-LC3, which was more striking for the S316A mutant compared with the S316D mutant (Fig. 3b, *upper panel*). In addition, the phospho-mimetic ATG4B S316D mutant reacted more strongly with the anti-pATG4B(Ser316) antibody as seen by western blotting (WB), confirming the ability to mimic Ser316 ATG4B phosphorylation (Fig. 3c). Importantly, we found that the expression of ATG4B S316D mutant displayed little catalytic activity against LC3 in the luciferase release assay compared with WT ATG4B (Fig. 3c). In fact, the activity of this mutant was comparable to the catalytic dead ATG4B C74S mutant. Notably, the ATG4B S316A mutant also showed a lower catalytic activity, attesting to the importance of this site but impeding the possibility of testing the effects of the non-phosphorylatable mutant on autophagy. To confirm that these effects were direct, we produced the recombinant phosphomimetic ATG4B S316D mutant in *Escherichia coli* and

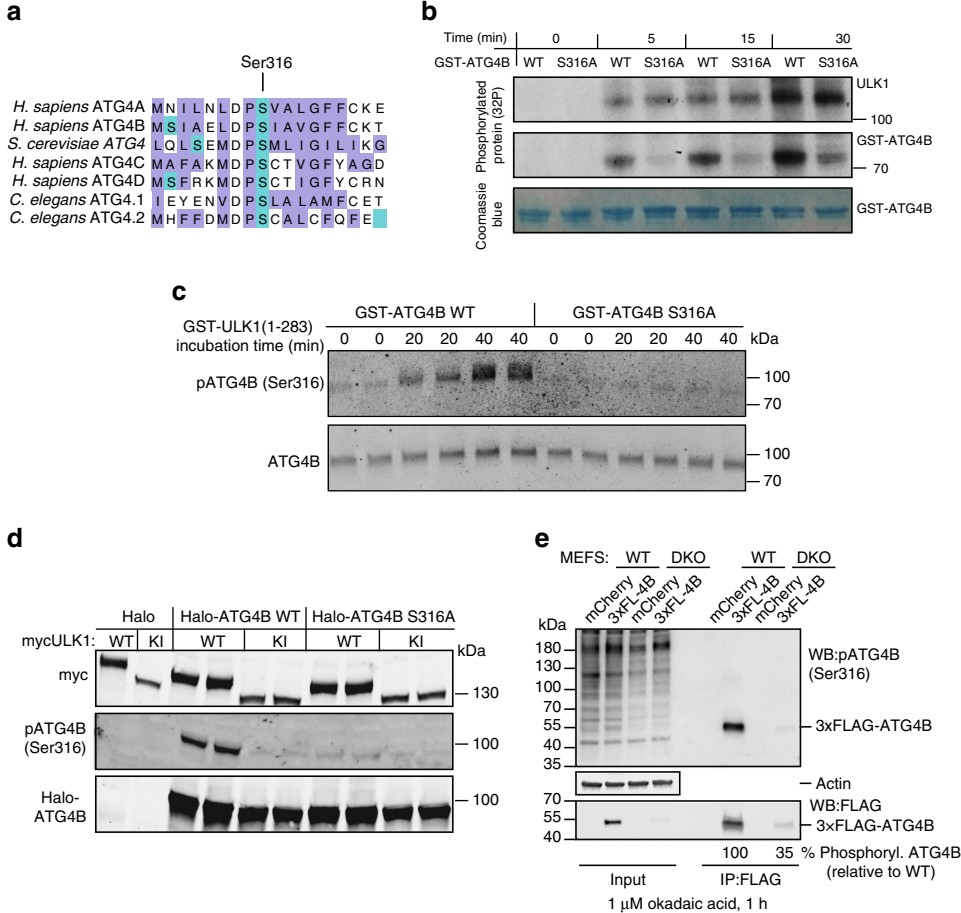

**Fig. 2** ATG4B Ser316 is a target residue of ULK1 phosphorylation. **a** Clustal Omega multiple alignment of ATG4 protein sequences surrounding hATG4B serine 316 from different isoforms and species; highlighted in *green* are serine residues and in *violet* hydrophobic amino acids. **b** In vitro radioactive phosphorylation assay with WT or ATG4B S316A mutant at different time points after addition of recombinant active ULK1. Coomassie blue staining is shown as an ATG4B loading control. **c** GST-ATG4B or GST-ATG4B S316A mutant was incubated for the indicated times with GST-ULK1 (1–283); samples were then assayed with a custom phospho-specific antibody against Ser316 of human ATG4B—pATG4B (Ser316) or total ATG4B. **d** Total lysates from wild-type (WT) or ULK1/2 double knockout (DKO) mouse embryonic fibroblasts (MEFs) transfected with 3×-FLAG-tagged ATG4B or mCherry as a control and treated with 1 μM okadaic acid for 1 h were probed with the pATG4B(Ser316) antibody and with an anti-ACTIN antibody as loading control. In parallel, lysates from the same samples were subjected to immunoprecipitation with FLAG M2 affinity gel and probed for pATG4B(Ser316) and FLAG antibody. Phosphorylation level was calculated using densitometry, with pATG4B(Ser316) signal divided by FLAG signal for the same band, expressed as a percentage of the WT MEF IP sample. **e** HEK293T lysates from samples co-transfected with mouse WT myc-ULK1 or Kinase Inactive (KI) myc-ULK1 and the indicated Halo constructs for 24 h were blotted and probed with the indicated antibodies (one representative blot from three independent experiments)

tested its activity in vitro towards the LC3B-GST construct. Compared with WT ATG4B, samples incubated with the phospho-mimetic mutant showed little free GST accumulation, again, almost as low as in the samples incubated with the catalytic dead ATG4B C74S mutant (Fig. 3d). To study whether LC3-PE de-lipidation activity was also affected by phosphorylation on Ser316 of ATG4B, we tested the ability of the phospho-mimetic S316D mutant to de-lipidate LC3-PE from membrane-enriched preparations. We separated cytoplasmic and membrane fractions of HEK293T cells treated with Torin1 (a potent mammalian target of rapamycin (mTOR) inhibitor) and bafilomycin A1 (a late-stage autophagy inhbitor) and incubated the samples with recombinant ATG4B WT, S316D and C74S mutants (Fig. 3e). The membrane fraction showed a large amount of lipidated LC3-II compared with the cytoplasmic fraction in treated cells, which was sensitive to de-lipidation by WT ATG4B. Interestingly, we found ATG4B S316D to be inactive against cleavage of lipidated LC3, similar to catalytic inactive ATG4B C74S (Fig. 3e), suggesting that the phospho-mimetic ATG4B S316D is inactive in processing of LC3-II.

**Phosphomimetic ATG4B S316D mutant reduces LC3 processing.** In order to assess the role of Ser316 phosphorylation on the formation of LC3-positive autophagosomes, we used ATG4B knockout cells generated by CRISPR/Cas9 genome editing for rescue experiments. The haploid HAP1 cell line was modified by CRISPR/Cas9 editing as described, resulting in a 16 nt deletion and premature termination stop codon in exon 4 (Fig. 4a). The absence of ATG4B in these cells was confirmed by WB (Fig. 4b). ATG4B knockout cells were then transfected with empty Halo vector as control, Halo-ATG4B, Halo-ATG4B C74S and Halo-ATG4B S316D and analysed for formation of LC3-positive puncta (Fig. 4c). Staining with Halo ligand dye revealed a variable level of construct expression (Fig. 4c). As ATG4B knockout cells are deficient in LC3 processing, we expect an absence of LC3-positive puncta in these cells. Indeed, in ATG4B knockout cells transfected with empty Halo-tag vector, no LC3-positive puncta could be observed upon co-treatment with Torin1 and bafilomycin A1, suggesting that ATG4B KO cells are unable to cleave pro-LC3 (Fig. 4c and Supplementary Fig. 3). In contrast, when WT ATG4B was expressed LC3 puncta could be observed,

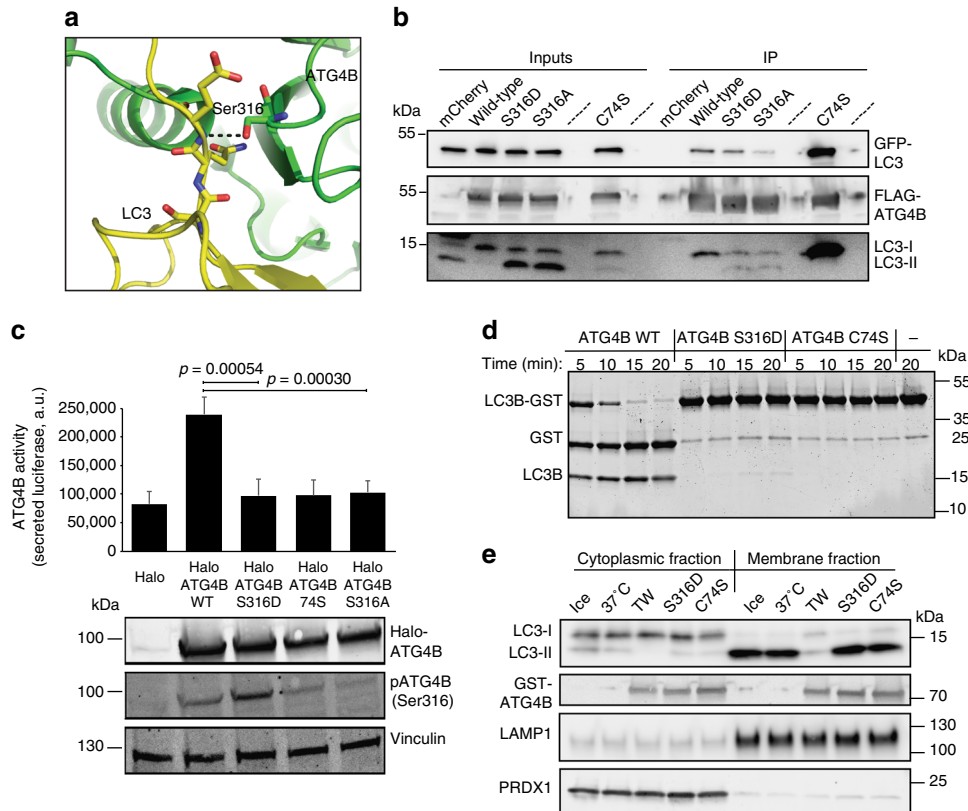

**Fig. 3** ATG4B Ser316 phosphorylation reduces LC3 binding and inhibits its catalytic activity towards unprocessed and lipidated LC3 in vitro and in cells. **a** Overview of the ATG4B-LC3 co-crystal structure from PDB 2Z0E[24], zooming at ATG4B Ser316 location. LC3 is in *yellow* and ATG4B is in *green*. *Red* and *blue* represent the oxygen and nitrogen atoms, respectively. Hydrogen bond is shown as a *black dashed line*. The molecular structure was generated with PyMOL. **b** HeLa cells stably expressing GFP-tagged LC3B were transfected with 3×FLAG-tagged ATG4B and ATG4B mutants or mCherry as control and subjected to immunoprecipitation with FLAG M2 affinity gel. The blots were probed for interaction of the 3×FLAG-tagged proteins with GFP-LC3B using GFP antibody (*upper panel*) and endogenous LC3B with LC3B antibody (*lower panel*), and 3×FLAG-ATG4B was detected using FLAG antibody (*middle panel*). **c** ATG4B activity was measured in the Actin-LC3-DelN reporter cell line 48 h after transfection with the WT, catalytic inactive (C74S), phosphomimetic (S316D) and non-phosphorylatable (S316A) Halo-ATG4B constructs ($n = 3$, average ± s.d., and representative blot showing the correct expression of the constructs). Values displayed are the relative intensity of phosphorylated ATG4B Ser316 normalized to Halo-ATG4B, relative to WT HaloATG4B expression (unpaired two-sided *t*-test). **d** Purified recombinant WT, phosphomimetic (S316D) and catalytic inactive (C74S) GST-ATG4B (0.004 mg mL$^{-1}$) were tested in vitro for their ability to cleave LC3B-GST (1 mg mL$^{-1}$). **e** Cytoplasmic and non-nuclear membrane fraction preparations from HEK293T cells treated with 250 nM Torin and 10 nM bafilomycin A1 for 3 h were incubated with the indicated GST-ATG4B WT, S316D and C74S mutants for 30 min at 37 °C, blotted and probed with the indicated antibodies. LAMP1 and PRDX1 were used as loading controls for the membrane and cytoplasmic fractions, respectively, and GST-ATG4B was detected using anti-ATG4B (one representative blot is shown)

which were increased in number upon Torin1 and bafilomycin A1 treatment, showing the recovery of ATG4B function and the ability to activate LC3 to enable its lipidation (Fig. 4c and Supplementary Fig. 3). Importantly, this occurred in cells that expressed low levels of Halo-ATG4B as those expressing higher levels showed no LC3 puncta formation likely due to excessive LC3-ATG4B interaction impeding LC3 function, as it has been previously observed[25]. Cells expressing Halo-ATG4B C74S showed no LC3 puncta regardless of expression level, consistent with an inability to process pro-LC3. In ATG4B knockout cells complemented with the phospho-mimetic Halo-ATG4B S316D mutant, LC3-positive puncta were observed showing that the remaining activity is sufficient to allow pro-LC3 cleavage. Unlike ATG4B knockout cells expressing ATG4B WT however, in ATG4B S316D-expressing cells even at the highest expression levels, LC3 puncta were still observed, proving that the phospho-mimetic mutant display defective LC3-PE de-lipidation or LC3 interaction also in vivo (Figs. 4c, d). Similarly, the S316A mutant also showed LC3 puncta in highly expressing cells, further supporting the notion that LC3 binding and processing is impaired in the ATG4B mutants. In conclusion, we propose a

model whereby ATG4B activity is suppressed by ULK1-mediated phosphorylation of Ser316. However, an ULK1-mediated inhibition of ATG4B poses a critical question: how is ATG4B re-activated to allow de-lipidation, which is considered to occur distal to ULK1-positive forming autophagosomes? One possibility is that a phosphatase exists that removes the inhibitory Ser316 phosphorylation, thus allowing for a reversible phospho-switch to control ATG4B enzymatic activity.

**PP2A is an endogenous regulator of ATG4B Ser316 dephosphorylation.** To identify such phosphatase, we screened a cDNA expression library including 74 phosphatases using the cell-based ATG4B luciferase release assay[20] (Fig. 5a and Supplementary Data File). We identified a number of protein phosphatases that enhanced ATG4B-mediated cleavage of LC3 when overexpressed, which were candidates for removing the phosphate group on ATG4B Ser316 in cells. We co-expressed the phosphatases that enhanced ATG4B-mediated LC3 processing in the presence of Halo-ATG4B and myc-ULK1 to test their ability to dephosphorylate ATG4B at serine 316 (Fig. 5b). Of the selected

phosphatases PPM1A, PPM1F and the regulatory subunit 3B of PP2A were found to decrease the phosphorylation status of ATG4B on serine 316 (Fig. 5b). Consistent with this finding, inhibition of PP2A with okadaic acid in cells stably expressing Halo-ATG4B, resulted in a strong increase of ATG4B

Ser316 phosphorylation, showing that PP2A is endogenously involved in the dephosphorylation of ATG4B on this residue (Fig. 5c). How the activity of this phosphatase is controlled during autophagosome initiation and formation requires further investigation.

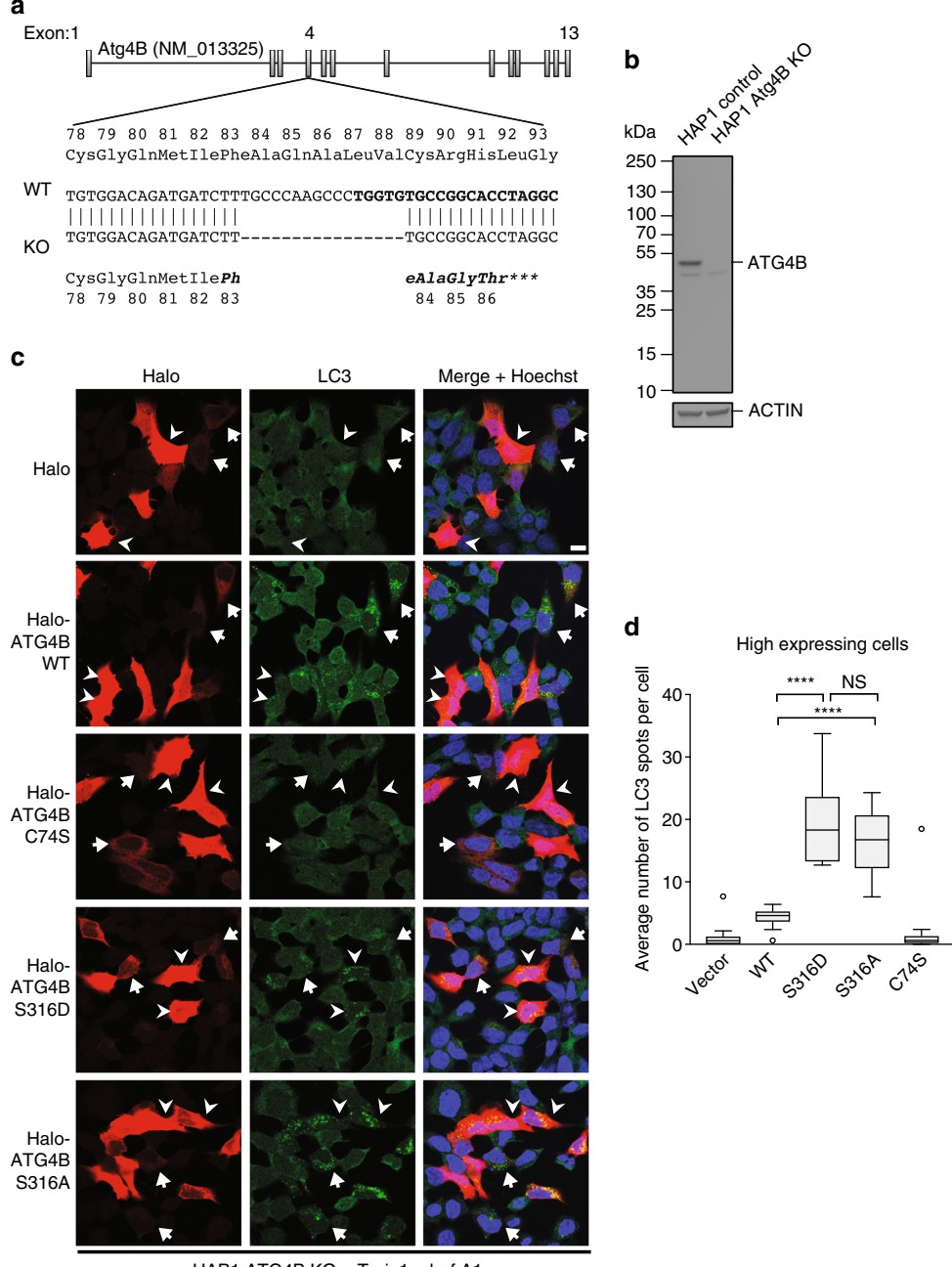

**Fig. 4** Quantification of LC3-positive autophagosomes in ATG4B knockout cells rescued with the phosphomimetic ATG4B S316D mutant. **a** Validation of CRISPR-mediated modification to the genomic locus of ATG4B exon 4 in the HAP1 ATG4B KO cell line. The gene structure of the WT human ATG4B locus on chromosome 2 is shown, and the region in exon 4 targeted for modification is revealed at the sequence level. The DNA sequence targeted by the custom sgRNA is shown in bold. The 16 bp deletion in the ATG4B KO clone results in a frameshift and early stop codon, as shown by the DNA sequence alignment. **b** Protein-level validation of ATG4B knockout. Lysates from untreated HAP1 control and ATG4B KO cells were run on a 4–20% polyacrylamide gel and immunoblotted with antibodies against ATG4B and beta-actin. The ATG4B KO cells showed a total absence of the specific band representing ATG4B. **c** ATG4B KO HAP1 cells were transfected with the indicated Halo constructs. After 24 h, cells were simultaneously stained live with a Halo ligand dye and treated with 250 nM Torin1 and 10 nM bafilomycin A1 for 3 h, prior to fixation and staining for endogenous LC3. *Arrowheads* indicate high Halo construct-expressing cells, and *arrows* indicate cells expressing low levels of these constructs. *Scale bar* 10 μm. **d** Quantification of the average number of autophagosome per cell in the high expressing cells is shown (Tukey's plot, $n = 10$ fields, with an average of 5 quantified cells per field). ****$p < 0.0001$, *NS* not significant (Sidak's multiple comparison test)

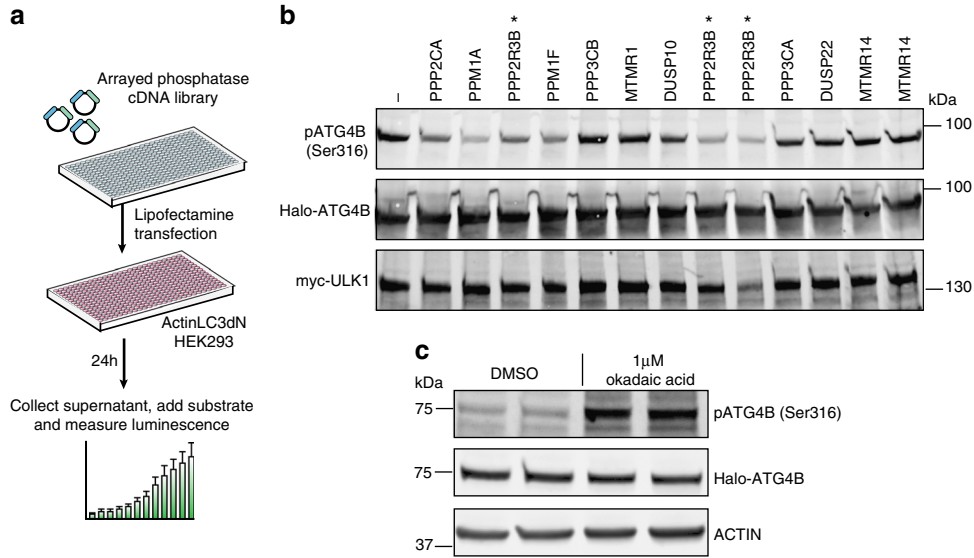

**Fig. 5** A phosphatase screen identifies PP2A as an endogenous regulator of ATG4B Ser316 dephosphorylation. **a** Schematic workflow of the cDNA phosphatase screen. **b** HEK293T cells after 24 h co-transfection with selected phosphatases and myc-ULK1 and Halo-ATG4B were analysed for the phosphorylation status of ATG4B on Ser316. The samples overexpressing the PP2A regulatory subunit 3B (PPP2R3B) are marked with an *asterisk*. Please note that the three different cDNA clones were transfected for PPP2R3B, all of which showed a reduction in phosphor-Ser316 compared with the control sample. Representative immunoblots are shown. **c** Cell lysates from HeLa cells stably expressing Halo-ATG4B WT and treated for 30 min with 1 μM okadaic acid or DMSO were probed for phosphorylation of ATG4B on Ser316 and then re-probed for Halo and β-actin expression. Representative immunoblot from three independent experiments

## Discussion

In summary, we have discovered that ULK1 inhibits LC3 processing through direct phosphorylation and inhibition of ATG4B. We have identified the target residue as Ser316, which lies at the interaction interface with LC3B and we propose that LC3 co-ordination is impaired through the bulky negative charged phospho-serine residue. We propose that LC3 processing is controlled by a reversible phospho-switch to inhibit and re-activate the catalytic activity of ATG4B, mediated by ULK1/2 and the opposing phosphatase. In fact, we have identified PP2A as a phosphatase able to dephosphorylate ATG4B on serine 316. PP2A is a family of >200 biochemically distinct heterotrimeric complexes composed of a combination of one catalytic domain, one scaffold subunit and one of many possible regulatory subunits[26]. The PP2A inhibitors okadaic acid and rifampicin have been shown to block starvation-induced autophagy[27–29] and specific PP2A-regulatory subunits have been linked to the regulation of autophagy by acting upstream on mTOR signalling. Our identification of PP2R3B-containing PP2A complexes in the regulation of ATG4B provides an mTOR-indepent molecular explanation for the described role of PP2A in autophagosome expansion observed in *Drosophila* and mammalian cells[30, 31]. However, the regulation of PP2A activity during autophagosome formation is complex and complicated by the fact that loss of PP2A activity modulates early steps in autophagosome formation. For instance, PP2A activity is required for dephosphorylation of ULK1 and activation of the ULK1 kinase[29]. Thus an inhibition of PP2A activity translates into a block of autophagy that is correlated with a reduction in ULK1 phosphorylation targets. Further studies are needed to distinguish between specific subsets of PP2A and their role in early and late steps in autophagy.

Although we do not demonstrate ULK1-dependent phosphorylation of endogenous ATG4B, we propose that cycles of phosphorylation and dephosphorylation of ATG4B regulate the activity of ATG4B and LC3 processing. Indeed, endogenous phosphorylation of S316 has been demonstrated in another study[22], thus supporting our findings. How this regulatory phospho-switch is controlled within the spatio-temporal context of autophagy is presently unclear. As ULK1 is mainly localized to the forming autophagosome upon induction of autophagy, it is possible that localized activation of ULK1 at the autophagosome formation site results in the phosphorylation of ATG4B on serine 316, thus protecting nascent autophagosomes from premature LC3 de-lipidation and, in concert with the conjugation machinery, increasing overall LC3-II availability. At the same time, inhibition of ATG4B-mediated de-lipidation at the immature autophagosomes ensures that LC3 coating is maintained through early steps in autophagosome formation. Over time, ULK1 may dissociate from autophagosomes or alternatively become inactivated by a yet unknown mechanism, thus releasing the inhibitory phosphorylation of ATG4B during later stages of autophagy. In addition, the presence of highly abundant PP2A might ensure dephosphorylation of ATG4B at sites of low ULK1 activity, thus allowing ATG4B to process pro-LC3 and potentially to de-lipidate incorrectly lipidated LC3–PE that can occur on non-autophagosome membranes[32]. It remains to be assessed whether the negative regulation exerted by oxidation on ATG4B activity[33] acts in concert or independently of ULK1 regulation. This could be aided by addditional post-translational modifications such as phosphorylation at Ser383 and Ser392 by as yet unidentified kinase or kinases that activates LC3 de-lipidation[34]. These studies pave the way to the understanding of how autophagy genes are regulated by post-translational mechanisms in order to allow the co-ordination of autophagosome formation and lysosomal delivery.

## Methods

**Plasmids.** Halo-ATG4B expression plasmid (pHTNATG4B) was created by sub-cloning full-length *ATG4B* from pEAK-ATG4B with *Eco*RI and *Not*I into pHTN-Halotag CMV-neo plasmid (Promega). Mutant S316D was generated by PCR using Ser316 forward (5′-GACCCGGACATCGCTGTGGG-3′) and reverse (5′-CCCACAGCGATGTCCGGGTC-3′) primers, C74S with forward (5′-GCTGGGGCAGCATGCTGC-3′) and reverse (5′-GCAGCATGCTGCCCCAGC-3′) primers and S316A with forward (5′-GACCCGGCCATCGCTGTGGG-3′) and reverse (5′-CCCACAGCGATGGCCGGGTC-3′) primers. pGEXATG4B was

created by sub-cloning *ATG4B* from pEAK-ATG4B with *Eco*RI and *Not*I into the bacterial expression vector pGEX6P-1 (GE Healthcare, 27-4597-01). pGEXATG4B$^{C74S}$ was created by PCR using pGEXATG4B as a template and primers C74S forward (5′-CCATGCTGCGGTGTGGACAGATG-3′) and C74S reverse (5′-AGCCCCAGCCTGTGTCCGAGGT-3′), pGEXATG4B$^{S316A}$ was created by PCR using pGEXATG4B as a template and primers S316A forward (5′-GCCATGCGCTGTGGGGTTTTTCTGT-3′) and S316A reverse (5′-CGGGTCA AGCTCCGCGATGCTCAT-3′), pGEXATG4B$^{S316D}$ created by sub-cloning *ATG4B S316D* from pHTNATG4B$^{S316D}$ with *Eco*RI and *Not*I into the bacterial expression vector pGEX6P-1. pGEX GST-ULK1cd, encoding *ULK1* catalytic domain (ULK1cd, amino acids 1–283), was obtained by PCR using full-length human *ULK1* as a template and EcoRI-ULK1cd forward (5′-AAAGAATTCATG GAGCCCGGCCGGCGGCACAG-3′) and ULK1cd-stop*Not*I reverse (5′-GCG GCCGCTTACGAGGGGCTGGCATCGAGGAAAGC-3′) primers and cloned into the bacterial expression vector pGEX6P-1 using *Eco*RI and *Not*I. pETLC3-GST was created by PCR amplifying *LC3B2* from the Actin-LC3-DelN reporter with LC3 forward (5′-ATGCCGTCGGAGAAGAACCTTCAAG-3′) and reverse (5′-CACTGA CAATTTCATCCCGAACGT-3′) primers and amplifying GST from pGEX6P-1 with GST forward (5′-ATGTCCCCTATACTAGGTTATTGG-3′) and GST-*Hin*dIII reverse (5′-TTTAAGCTTAGGTTTTCACCGTCATCACCGAAA-3′) primers and the resulting fragments were cloned in pET21d (Novagen, 69743-3) previously digested with *Nco*I, blunted by Klenow large fragment DNA polymerase and *Hind*III. All PCR were performed at 30 cycles using either Pyrobest DNA polymerase (Takara, R005A) (For pGEXATG4B$^{C74S}$, pGEXATG4B$^{S316A}$, pGEX GST-ULK1cd and pET LC3-GST) or Phusion polymerase (New England Biolabs, M0530S). pDONR223 ATG4B was created by recombining *ATG4B* amplified by PCR from pHTN-HaloTag-ATG4B with pDONR223 and transforming in DH5alpha and selecting on Ampicillin/Spectinomycin plates. A lentiviral vector for the expression of N-terminally Halo tagged proteins was created by cloning HaloTag from pHTN-HaloTag vector in the *Eco*RV and *Nhe*I sites of the lentiviral pLD-Gateway-Puro-NVF (gift of Julia Petschnigg) and used to create the gateway compatible N-terminal HaloTag-ATG4B vector with a Neomycin mammalian selection cassette. 3×FLAG-tagged ATG4B mutant expression vectors were generated by gateway recombination of ATG4B entry clones with pCMV-tripleFLAG-Gateway[35]. Mouse Myc-ULK1 and Myc-ULK1 kinase inactive K46I mutant plasmids were a kind gift from Sharon Tooze, and myc-hULK1 was a gift from Do-Hyung Kim (Addgene plasmid no. 31961)[36].

**Chemical probes.** Chemical probes' stock solutions were made in dimethyl sulphoxide at a concentration of 10 μM bafilomycin (Sigma, B1793), 250 μM Torin1 (Merck-Millipore, 475991) and 1 mM okadaic acid (Cell Signaling, 5934S). Halo ligand dye used was HaloTag TMRDirect Ligand (Promega, G2991) and added to the culture medium of live cells at a final concentration of 0.1 μM for the same duration as compound treatment.

**Antibodies.** A rabbit polyclonal phospho-serine 316 ATG4B antibody was custom produced by GeneScript by rabbit immunization. pATG4B(Ser316) was diluted 1:1000 in PBST (PBS 0.005% Tween-20) 3% bovine serum albumin (BSA, Sigma-Aldrich, A7906).

Commercially available antibodies and dilutions used are s follows: rabbit ATG4B antibody (Cell Signaling, 5299S, WB dilution 1:1000); mouse monoclonal antibody Anti-HaloTag (Promega, G9211, WB dilution 1:1000); rabbit anti-LC3B antibody (Sigma-Aldrich, L7543, WB dilution 1:1000, immunofluorescence dilution 1:400); rabbit anti-Peroxiredoxin 1 antibody (ABfrontier, LF-MA0031, 1:1000); mouse anti-β-actin (Sigma A1978, WB dilution 1:2000); mouse anti Myc (Millipore, CB430, WB dilution 1:400); rabbit anti-Vinculin (Abcam, AB129002, WB dilution 1:10000); mouse anti-LAMP1 (BD Biosciences, 611042, WB dilution 1:500); mouse anti-GFP (Clontech, 632381, WB dilution 1:1000); mouse biotinylated anti-FLAG (Sigma-Aldrich, F9291, WB dilution 1:1000); and goat anti-GST (GE Healthcare, 27-4577-01, WB dilution 1:1000).

**Protein purification.** Expression plasmids pGEXATG4B, pGEXATG4B$^{C74S}$, pGEXATG4B$^{S316A}$, pGEXATG4B$^{S316D}$, pGEXULK1cd and pETLC3B-GST were transformed into *E. coli* BL21 (DE3)-R3-lambda-PPase obtained from the Structural Genomics Consortium, Oxford University, Oxford, UK. The proteins were induced by 0.1 mM isopropyl-D-1-thiogalactopyronoside at 18 °C overnight. GST-tagged proteins were purified using Glutathione Sepharose 4B (GE Health-care, 17-0756-01) as described previously[37]. The proteins were further purified and concentrated using Amicon ultra centrifulgal filter units ultra-15, molecular weight cutoff (MWCO) 30 kDa (Sigma-Aldrich, Z717185-8EA) or MWCO 10 kDa (Sigma-Aldrich, Z706345-8EA). PreScission Protease (GE Healthcare, 27-0843-01) was used for digestion when necessary to remove GST from GST-tag proteins. The following recombinant proteins were commercially obtained: GST-tagged ULK1 (Sigma, SRP5096) and GST-tagged ATG4A (Abnova, H00115201-P01). All recombinant proteins were stored at −80 °C in 50 mM Tris-HCl, pH 8.0, 150 mM NaCl, 0.5 mM EDTA, 0.1 mM EGTA, 33% glycerol and 1 mM dithiothreitol (DTT). Sequence alignment was performed with Clustal Omega[38] at www.uniprot. org. Protein structure visualizations were generated with PyMOL Molecular Graphics System (DeLano Scientific, Palo Alto, CA, USA).

**In vitro phosphorylation assays.** In vitro radioactive assays were performed by incubating 100 ng recombinant ATG4B diluted in assay buffer (20 mM Tris-HCl pH 7.5, 10 mM MgCl$_2$, 5 mM DTT, 20 μM cold ATP and 0.16 μM ATP [γ-32P] PerkinElmer NEG502A100UC) in the presence of 10 ng recombinant ULK1 (catalytic domain) or 3 μL of active recombinant ULK1 (Sigma-Aldrich, SRP5096) at 30 °C for the indicated times. The reaction was stopped by adding 5× sample buffer (250 mM Tris-Cl pH 6.8, 10% sodium dodecyl sulphate (SDS), 50% glycerol, 25% β-mercaptoethanol or 0.5 M DTT and 0.05% bromophenol blue) and boiling for 5 min. Samples were loaded on NUPAGE Acrylamide gel (Invitrogen, NP0321BOX). Gels were stained with InstantBlue Protein Stain (Expedeon, ISB1L) before drying on filter paper and measuring incorporated radioactivity by exposing on photographic film (Bio-Rad).

**LC3-GST assay.** LC3-GST was diluted at 1 mg mL$^{-1}$ in assay buffer (50 mM Tris-HCl pH 8.0, 150 mM NaCl, 5 mM DTT), and after addition of ATG4B (0.005 mg mL$^{-1}$), samples were incubated at 37 °C and the reaction was stopped by adding 5× sample buffer and boiling for 5 min. For testing the effects of ULK1 on ATG4B activity, the LC3-GST assay was performed in the Kinase Buffer as described. For testing the dependency of the reaction to ATP, 2 μg of recombinant WT ATG4B or ATG4B C74S was incubated with 0.02 μg ULK1 in the presence or absence of 1 mM ATP for 30 min at 37 °C. ATG4B and the C74S mutant were then diluted 500× and incubated with LC3-GST and incubated for 60 at 37 °C. Samples were loaded on Mini-PROTEAN TGX Precast Gels (Bio-Rad) and run on Bio-Rad mini-protean tetra electrophoresis system (Bio-Rad). Gels were stained in InstantBlue Protein Stain (Expedeon, ISB1L) and imaged on a LiCor Odyssey fluorescence scanner at a wavelength of 700 nm. Images were quantified in Fiji[39] (http://fiji.sc/Fiji) using the analyse>gel built-in function.

**LC3-PE enrichment.** HEK-293T cells were collected and washed in PBS. Cells were then homogenized in Resuspension Buffer (Hepes-KOH pH 7.5, 0.22 M Mannitol, 0.07 M Sucrose and protease inhibitors) by passing the sample 10 times through a 27 G needle. Nuclei were discarded by centrifugation at 300×g for 10 min, and the cytosolic and membrane fraction were separated by centrifugation at 7000 g for 10 min. The membrane pellet was resuspended in the same volume of resuspension buffer and 0.5% Triton X100 and 5 mM DTT were added to each fraction. Recombinant ATG4B was then added at a concentration of 0.004 mg mL$^{-1}$ and incubated at 37 °C for 20 min, after which 5× sample buffer was added and samples were boiled at 95 °C for 5 min and assessed by WB.

**Cell lines culture and transfection.** MEF cells and short tandem repeat (STR)-profiled HeLa (ATCC_CCL2) and HEK293T (ATCC_CRL-3216) cells were cultured in Dulbecco's modified Eagle's medium (DMEM; Life Technologies, 61965) supplemented with 10% foetal bovine serum (FBS; Life Technologies, 10500), 10 U mL$^{-1}$ Penicillin and 0.1 mg mL$^{-1}$ Streptomycin (Sigma-Aldrich, P0781) and 1 mM Pyruvate (Life Technologies, 11360) at 37 °C 5% CO$_2$. HAP1 cells were cultured in Iscove's Modified Dulbecco's Medium (IMDM; Life Technologies, 21980-065) supplemented with 10% FBS and antibiotics at 37 °C 5% CO$_2$. HAP1 ATG4B knockout cells were generated by and obtained from Horizon Genomics (HZGHC001241c011). ULK1/2 DKO MEFs were obtained from Sharon Tooze.

MEF, HEK293T and HeLa cells were transfected with PEI "Max", MW 40,000 (Polyscience, 24765-2). PEI was added at a ratio of DNA (μg):PEI (μg) of 1:4 to DNA diluted in DMEM without serum and antibiotics, briefly vortexed and added to cells growing in complete medium after a 20 min incubation. After 4–16 h, the medium was replaced with fresh medium. HAP1 cells were transfected with X-tremeGENE HP (Roche, 06366244001) at a 1:1 ratio with 500 ng of DNA in 24-well plates, changing medium after 4 h.

**Lentiviral packaging and transduction.** Halo was cloned in a lentiviral Gateway destination vector, under the control of a CMV promoter and harbouring a mammalian Neomycin selection cassette. HEK293T cells were grown in complete DMEM in a six-well plate and co-transfected with 900 ng psPAX2 (packaging plasmid, Addgene no. 12260), 100 ng pMD2.G (envelope plasmid, Addgene no. 12259) and 1 μg of Halo-ATG4B lentiviral plasmid per well using 2 μL X-tremeGENE per μg of DNA. At 18 h post-transfection, medium was replaced with complete IMDM. Viral supernatant was harvested at 72 h post-transfection, 0.22 μm filtered and frozen at −80 °C. HeLa cells were seeded in 24-well plates and infected overnight with 300 μL of viral supernatant containing 8 μg μL$^{-1}$ polybrene. Cells were selected 48 h postinfection for 1 week in the presence of 400 μg mL$^{-1}$ G418 (Sigma-Aldrich, no. G8168).

**ATG4B activity cell assay.** STR-profiled HEK293T cells stably expressing the Actin-LC3-DelN reporter were seeded in 96 multiwell plates at 20,000 cells per well. A total of 250 ng of DNA were transfected by mixing with 0.7 μL of Lipofectamine 2000 (Life Technologies, 11668-019) in a 50 μL total volume of OPTIMEM (Life Technologies, 31985), incubating 20 min at room temperature (RT) and then adding to the wells. After 48 h, 10 μL of supernatant were transferred to a new plate (Greiner Bio-One, 655076) and 90 μL of 10 μg mL$^{-1}$ native coelenterazine (Insight Biotechnology, sc-205904, stock 1 mg mL$^{-1}$ in MeOH/1%

HCl) diluted in 0.5 M Tris-HCl pH 7.4 was added prior to measuring luminescence in a PerkinElmer Envision II plate reader.

**Immunocytochemistry and confocal microscopy.** HAP1 ATG4B knockout cells were grown on glass coverslips, fixed with 4% paraformaldehyde for 15 min RT, washed twice in PBS, fixed in ice-cold MeOH for 15 min at −20 °C, washed twice in PBS and quenched 20 min at RT in 50 mM NH$_4$Cl. After permeabilization with 0.1% TX-100, samples were blocked in PBS 3% goat serum for 60 min. Primary and secondary antibodies were diluted in blocking solution and incubated for 60 min each at RT. Nuclei were stained with 1 µg mL$^{-1}$ Hoechst 33342 in PBS for 10 min. Coverslips were mounted with ProLong Diamond Antifade Mountant (Thermo-Fisher, P36970) and sealed with nail polish before imaging. Slides were imaged using a Leica TCS SP5 confocal laser scanning microscope (×63 oil immersion objective, NA = 1.4) with sequential channel acquisition. For discrimination of Halo-ATG4B expression, non-saturated images were used for quantification.

**Image quantification.** Confocal images were imported in Columbus 2.4 with Omero 4.4.7 (PerkinElmer). Imported images were then batch analysed with Acapella 3.1, with the following script: find nuclei in DAPI channel (method B), find cytoplasm in LC3 channel (method A), find spots in LC3 channel (method B, $>6px^2$), divide cells according to Halo mean cytoplasmic pixel intensity in high (>180), medium (>25, <180) and low (<25) and calculate the average number of LC3 spots in each class.

**Western blotting.** Cells were collected and washed once in PBS. Cells were then lysed in 1% NP-40 lysis buffer (50 mM Tris-HCl pH 7.5, 150 mM NaCl, 1% NP-40) supplemented with protease inhibitors (Complete EDTA-Free, Roche, cat 04693132001) and phosphatase inhibitors (PhosSTOP, Roche, cat 04906845001) for 15 min on ice and the insoluble material was discarded after 15 min centrifugation at 4 °C, 5× sample buffer was then added to the supernatant, boiled for 5 min and loaded on gels with PageRuler Plus Prestained (ThermoFisher, 26619) as protein ladder. For all WB, 4–20% or 7.5% precast Bio-Rad gels were used on a Bio-Rad mini-PROTEAN system and blotted on nitrocellulose 0.45 µm (Bio-Rad, 162-0115) or Immobilon-FL 0.45 µm polyvinylidene difluoride membrane (Merck-Millipore, IPFL00010) following the manufacturers' instructions. Membranes were blocked in PBS-T (PBS 0.005% Tween-20) 5% Milk (Sigma, 70166-500G) or TBS-T 3% BSA for phospho-specfic antibodies. Primary antibodies were diluted in blocking buffer and incubated for either 1 h at RT or 4 °C overnight. Secondary antibodies were goat anti-rabbit immunoglobulin G (IgG) (H+L) IRDye800 (LiCor, 926-32211) in PBS-T supplemented with 5% milk or TBS-T supplemented with 3% BSA and goat anti-mouse IgG (H+L) IRDye680 LT (LI-COR, 926-68020) in PBS-T supplemented with 0.02% SDS and 5% milk. Detection was performed on ImageQuant chemilumenscent imaging system (GE Healthcare) or Odyssey infrared imaging system (LiCor). For stripping antibodies from membranes, Restore PLUS Western Blot Stripping Buffer (ThermoScientific, 46430) was used following the manufacturer's instructions. Densitometry was performed as described for LC3-GST assay. Uncropped scans of the blots are provided in Supplementary Fig. 4.

**IP experiments.** HeLa cells stably expressing GFP-LC3B grown in 10 cm tissue culture dishes were transfected with cDNA expression constructs using PEI (a total of 10 µg DNA mixed with 40 µg PEI per dish). After 24 h, cells were washed in PBS and lysed in 600 µL of IP buffer (50 mM Tris-HCl pH 8, 150 mM NaCl, 1 mM EDTA, 1% Triton X-100 and protease inhibitors) and samples were cleared by centrifugation at 15, 000×g for 10 min at 4 °C. After 50 µL input sample was collected into a separate tube and boiled in sample buffer, the remaining cleared sample was loaded onto ANTI-FLAG M2 Affinity Gel (Sigma, A2220) that had previously been prepared by washing three times with 1 mL IP buffer. Approximately 30 µL of packed gel was used per sample, and binding was performed at 4 °C for 3 h on a tube rotator. Gel was washed three times with 1 mL IP buffer and bound protein was eluted from the gel by boiling in 60 µL sample buffer for 5 min at 95 °C. WB was performed as described and GFP was specifically detected with anti-GFP antibody and revealed using horseradish peroxidase (HRP)-conjugated protein A (Abcam, ab7456) to avoid detection of denatured mouse FLAG M2 antibody chains. 3×FLAG-ATG4B was subsequently specifically detected using biotinylated anti-flag antibody and revealed with streptavidin-conjugated IRDye680LT (LiCor, 926-68031) to avoid detection of mouse anti-GFP. Endogenous LC3B was specifically detected on a cut portion of the same membrane using goat anti-rabbit HRP (Santa Cruz, sc-2004).

For IP of 3×FLAG-ATG4B in WT and ULK1/2 DKO MEFs, a similar procedure was performed with modifications. At 48 h post-transfection, cells were treated for 1 h with 1 µM okadaic acid, washed in PBS and lysed in 1 mL MEF IP buffer (as above, but supplemented with phosSTOP and without EDTA). Following column binding and washing (once in MEF IP buffer and twice in MEF IP buffer lacking phosSTOP), competitive elution was performed in 120 µL MEF IP buffer lacking phosSTOP and containing 3×FLAG peptide (Sigma, F4799) at a concentration of 100 µg mL$^{-1}$. WB was performed and the entire membrane was first probed with pATG4B (Ser316). Following this, total 3×FLAG-ATG4B was detected using streptavidin-conjugated fluorescent secondary antibody as above,

and actin was detected using IRDye800CW goat anti-mouse secondary antibody (LiCor, 925-32210).

**Phosphatase treatment of cell extracts.** Cell lysates or IP samples were divided equally into tubes on ice containing 1/10 volume of Calf Intestinal Phosphatase (New England BioLabs, M0290L) or an equivalent volume of lysis buffer or IP buffer for control reactions. Tubes were incubated at 37 °C for the times indicated, and the reaction was stopped by addition of 5× sample buffer with immediate boiling at 95 °C. Samples were then subjected to WB analysis.

**Phosphatase cDNA expression screen.** HEK293T cells were transiently co-transfected with Actin-LC3-DelN and an arrayed cDNA expression library (OpenBiosystems, MHS5004_PhosFLcDNA_CMV) in triplicates. In each well of a 96-well plate, 30,000 cells were seeded and transfected the following day with 100 ng of the Actin-LC3-DelN reporter and 100 ng of each cDNA expression clone using Lipofectamine 2000. Supernatants were removed from cells after 24 h and assayed for luciferase activity as described previously[40]: 10 µL of supernatant were transferred to a new plate (Greiner Bio-One, 655076) and 90 µL of the 10 µg mL$^{-1}$ native coelenterazine (Insight Biotechnology, sc-205904, stock 1 mg mL$^{-1}$ in MeOH/1%HCl) diluted in 0.5 M Tris-HCl pH 7.4 was added prior to measuring luminescence in a PerkinElmer Envision II plate reader.

**Statistical analysis.** Indicated p-values were calculated using an unpaired two-tailed t-test in Microsoft Excel. Histograms show means and error bars represent s.d. Box plot in Fig. 4d was generated using the Tukey's method in Prism 7 (GraphPad) and p-values in this chart were calculated using ordinary one-way analysis of variance with Sidak's multiple comparisons test.

**Data availability.** All relevant data from this article are available from the authors.

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

## Acknowledgements

We thank Sharon Tooze for sharing plasmids and the ULK1/2 knockout fibroblasts and the Structural Genomics Consortium, Oxford University, Oxford, UK for the BL21 bacterial strain. STR profiling was carried out by Dr Volpi's group at the University of Westminster as part of a 'Cell Authentication' initiative for best laboratory practice kindly sponsored by the Faculty of Science and Technology. A special thank you to Britta Diedrich and Joern Dengjel as well as Christin Luft, Janos Kriston-Vizi, Joana Costa, Jack Heintze and Julia Petschnigg for support and advice. We also thank Melanie Weber and Katie Kelly for technical assistance. This work was supported by the UK Medical Research Council core funding to the MRC-UCL University Unit (Ref. MC_U12266B), BBSRC funding (BB/J015881/1) and the UCL CiC funding scheme supported by the Medical Research Council (MC_PC_12024).

## Author contributions

R.K. and N.P. conceived the study and wrote the manuscript. All authors designed and performed experiments and revised the manuscript. R.K. secured funding.

## Additional information

**Competing interests:** The authors declare no competing financial interests.

