## [Peer Review File · Nature Communications]

Reviewers' comments:

Reviewer #1 (Remarks to the Author):

This manuscript by Pengo et al addresses how the functions of ATG4 in LC3 processing are regulated by the ULK1 kinase. They show in vivo and in vitro that ULK1 phosphorylates ATG4B at Ser316. Phosphomimetic mutant S316D shows defective LC3 processing in an in vivo luciferase release assay and an in vitro assay measuring recombinant S316D cleavage of an LC3B-GST construct. To measure effects on delipidation, the authors use membrane fractions enriched with lipidated LC3-II, and find that LC3-II is cleaved by only wild-type ATG4 and not the phosphomimetic. Notably, when they transfected ATG4 knockout cells with the phosphomimetic, they observed LC3 puncta formation at high levels of S316D expression. Finally, they identify PP2A as a candidate phosphatase that might cooperate with ULK1 in the phospho-switch regulating ATG4 activity.

The paper makes the important point that ULK1 kinase phosphorylates ATG4B and that this seems to regulate its processing of LC3. However, there are some key issues that should be addressed before publication.

1) It is interesting that the nonphosphorylatable mutant S316A also shows a low catalytic activity in the luciferase release assay (Fig. 3a), essentially equivalent to the loss in catalytic activity shown in the the phosphomimetic mutant. This suggests that phosphorylation status at this residue is not the only factor responsible for the loss of catalytic activity. Alternatively, the hydrogen bond between Ser316 and a main chain nitrogen of LC3 could be important for affinity. The authors should address this and test the phosphonull mutant further – do the phosphomimetic and phosphonull mutants show similar phenotypes consistently across different assays? Particularly informative would be transfecting the Atg4 knockout cells with the S316A mutant and observing whether LC3 puncta are formed.

2) In the Atg4 knockout cell rescue experiment, it is notable that the phosphomimetic shows significant rescue, in contrast to the 74S-transfected cells, even though the phosphomimetic shows a loss in catalytic activity similar to that of the catalytic dead 74S mutant in several other assays. Is this a full rescue - how does the number of LC3 puncta in the high-expressing S316D cells compare to that in the low-expressing WT cells? What might account for the difference in phenotype between the S316D and 74S mutants only in this particular assay? Perhaps the authors could consider conducting another assay (eg, p62 accumulation) to measure effects of these mutants on autophagy.

3) The authors point out in Fig. 2 that Ser316 is at the interface between ATG4B and LC3 and might be involved in hydrogen bonding between these two proteins. However, they only show data on how phosphorylation affects catalytic activity and not on how it affects complex formation. It would strengthen the paper if the authors could also investigate changes in affinity (and include the phosphonull mutant as well, which would be predicted to decrease affinity by eliminating a hydrogen bond).

4) Minor point. It would be helpful if the authors could indicate what band is specifically being detected by the pATG4B antibody in the cell lysate western blot (Fig 2d).

Reviewer #2 (Remarks to the Author):

Autophagosome formation is a central event during autophagy and involves the ubiquitin-like protein Atg8/LC3. Atg4 is a cysteine protease responsible for C-terminal processing/delipidation of Atg8/LC3, and thus crucial for the regulation of autophagy. In this study, Pengo et al. found that

ATG4B can be phosphorylated by the autophagy-related kinase ULK1. The authors further identified a Ser residue subjected to phosphorylation by ULK1. Mutations at this Ser residue abolished ATG4B activity, probably because it is directly involved in the interaction with LC3 as seen in the previously-reported crystal structure of the ATG4B-LC3 complex. Based on these results, the authors propose that ULK1-mediated phosphorylation downregulates ATG4B activity to prevent delipidation of LC3, resulting in the promotion of autophagosome formation. Finally, the authors add data suggesting that PP2A phosphatase counteracts ULK1 to reactivate ATG4B. In summary, phosphoregulation of ATG4B is novel, and the model proposed is of potential interest. However, phosphorylation of ATG4B by ULK1 in the cell was shown only using cells overexpressing these proteins; observing ULK1-dependent phosphorylation of endogenous ATG4B is necessary. Moreover, the significance of ATG4B phosphorylation in autophagosome formation has not been demonstrated. Therefore, the data presented in the present version of the manuscript do not substantially support their conclusions.

Specific comments:

(1) To examine the significance of ATG4B phosphorylation by ULK1 in autophagosome formation, the authors need to obtain a non-phosphorylatable mutant of ATG4B. I wonder if the replacement of Ser316 with other residues such as Thr could retain the interaction with LC3. Alternatively, mutations in residues around Ser316, a consensus motif for phosphorylation by ULK1, may impair phosphorylation without affecting the interaction with LC3.

(2) The experiments in Figure 1b is very important to show that ULK1 modulates ATG4B activity, but contain some unclear points. Why are cleaved GST and LC3 fragments observed already at time 0? Whereas the authors claim that less LC3 was generated in the presence of ULK1, the level of GST fragments appears the same between the reactions with and without ULK1. On the other hand, the authors should put a control, in which a kinase dead mutant of ULK1 is added.

(3) Does the result shown in Figure 2d mean that this antibody broadly reacts with ULK1 substrates not specifically to ATG4B phosphorylated at Ser316?

(4) In Figure 3a, the authors should quantify the bands and show the values for phosphorylated ATG4B corrected using the intensities of the Halo-ATG4B bands.

(5) Whereas the author reason that loss of LC3 puncta in cells highly expressing ATG4B is due to excessive delipidation, Yoshimori's group reported that overexpression of ATG4B inhibits lipidation of LC3 (Fujita et al, 2008, MBoC).

(6) In the description on Figure 5b, it is unclear which protein corresponds to the regulatory 3B subunit of PP2A.

(7) It would be important to test if ULK1 also phosphorylates other ATG4 paralogs.

(8) In the text, the authors use the words autophagosome maturation probably to mean a late stage of autophagosome formation, but this would be confusing or misleading, since autophagosome maturation can be regarded to mean fusion of complete autophagosomes with endosomes/lysosomes.

Reviewer #3 (Remarks to the Author):

The mechanism by which ATG4 activity in priming and de-lipidation of LC3 conjugated to PE is regulated remains largely elusive. In their study Pengo et al. link phosphorylation of ATG4 Ser316

by ULK1 to inhibition of LC3II delipidation. Furthermore, the phosphatase PP2A is proposed to alleviate such inhibition. Using over expression and in-vitro assays ULK1 was shown to inhibit the function of ATG4. Further bioinformatics approaches and site-directed mutations identified Ser 316 as the residue that undergoes phosphorylation by ULK1. This phosphorylation was shown to inhibit cleavage of LC3B-GST in-vitro. Finally, preliminary evidence is provided to support the notion that Ser316 indeed regulates autophagy in tissue culture cells.

This work addresses interesting issue regarding the regulation of ATG4 function. However, the data are too preliminary to support the authors' model. It remains unclear whether phosphorylation of ATG4 affects autophagy in vivo. A more rigorous approach using different autophagic markers should be taken to determine whether autophagic flux is affected by Atg4 phosphorylation and at what stage along the process. Also, it is not clear under which physiological conditions such phosphorylation occurs and what triggers the action of the phosphatases.

Additional comments:

Figure 1A – mycULK1 KI is expressed to a lower extent compared to the wt, which might affect the activity of ATG4B.

Figure 1B – LC3-GST is over exposed and it appears that its level is different between the no ULK1 and the ULK1 lanes.

Figure 2C – there is lower level of ATG4B in the S316A construct compared to the wt which might affect the phosphorylation level.

Figure 2D – ATG4 should be immunoprecipitated prior to testing the anti-p316 antibody to show that indeed the phosphorylated form of ATG4 is recognized.

Figure 3C- loading control for the membranal fraction is needed.

Figure 4C – over expression of the S316D mutant looks similar to the wt control. The differentiation to high and low expressing cells is not convincing as in the figure there are highly expressing cells with few/many spots in both wt and the mutant over expression. Furthermore, untreated control should be also presented in the figure.

Figure 5C – Autophagic flux should be also examined in the presence of okadaic acid. The specific effect of okadaic acid on PP2A should be demonstrated and more specific genetic approaches should be taken to identify the relevant phosphatases.

Reviewer #1 (Remarks to the Author):

- 1) *It is interesting that the nonphosphorylatable mutant S316A also shows a low catalytic activity in the luciferase release assay (Fig. 3a), essentially equivalent to the loss in catalytic activity shown in the phosphomimetic mutant. This suggests that phosphorylation status at this residue is not the only factor responsible for the loss of catalytic activity. Alternatively, the hydrogen bond between Ser316 and a main chain nitrogen of LC3 could be important for affinity. The authors should address this and test the phosphonull mutant further – do the phosphomimetic and phosphonull mutants show similar phenotypes consistently across different assays? Particularly informative would be transfecting the Atg4 knockout cells with the S316A mutant and observing whether LC3 puncta are formed.*

Reply:

We have performed the experiment as suggested and investigated the effect of the S316A mutant on LC3 puncta formation. The S316A mutant behaved similar to the S316D mutant, showing an increase of LC3-positive puncta in cells. This result supports the observation that both S316D and S316A have a defect in catalytic processing, which – as the reviewer suggests and we have pointed out in the manuscript – can be due to a decreased coordination of LC3 with ATG4B. To address this further and as the reviewers have suggested, we have performed co-immunoprecipitation experiments of ATG4B and ATG4B mutants with LC3. Indeed, we see reduced binding of LC3 to both S316D and S316A, indicating that binding might be impaired by modification of this residue. The new experiment is shown as new Figures 3b (Co-IP) and 4c (Imaging of S316A in 4B knockout cells).

- 2) *In the Atg4 knockout cell rescue experiment, it is notable that the phosphomimetic shows significant rescue, in contrast to the 74S-transfected cells, even though the phosphomimetic shows a loss in catalytic activity similar to that of the catalytic dead 74S mutant in several other assays. Is this a full rescue - how does the number of LC3 puncta in the high-expressing S316D cells compare to that in the low-expressing WT cells? What might account for the difference in phenotype between the S316D and 74S mutants only in this particular assay?*

Reply:

This is an excellent point. Before doing this experiment, our expectation was exactly the same as the one raised by the reviewer and we were intrigued to find that S316D cells displayed puncta in the first place.

However, C74S and S316D show reduced ATG4B activity for different reasons: C74S is a mutation in the catalytic site, leading to complete absence of activity, whereas S316D reduces activity rather than completely abolishes it (as shown in Fig. 3d). As a consequence, S316D has some residual activity that is sufficient to activate the pro-LC3 cleavage, thus leading to puncta formation over time. This is particularly important when there are high levels of S316D expressed, leading to formation of puncta, but possibly reduced de-lipidation. Another difference between C74S and S316D is that C74S strongly binds to LC3, whereas S316D shows reduced binding as shown in the co-IP experiment that we added to the manuscript (Fig. 3b). This could additionally impact the formation of LC3-positive structures.

Nonetheless, the quantification shows a striking difference for C74S as well as S316D when compared to the wild-type ATG4B control, supporting that both are indeed inhibitory to overall

activity. Also, we should point out that selection of high- vs low-expressing cells is done in an unbiased manner by using the threshold function of the image analysis software. Therefore, the results came out as they are and are not in any way biased by the experimenter.

- 3) *The authors point out in Fig. 2 that Ser316 is at the interface between ATG4B and LC3 and might be involved in hydrogen bonding between these two proteins. However, they only show data on how phosphorylation affects catalytic activity and not on how it affects complex formation. It would strengthen the paper if the authors could also investigate changes in affinity (and include the phosphonull mutant as well, which would be predicted to decrease affinity by eliminating a hydrogen bond).*

Reply:

We have performed co-immunoprecipitation experiments to show binding of LC3 to ATG4B and the phospho-mutants. In agreement with the overall manuscript, we observed less binding of endogenous LC3 to ATG4B S316A as well as S316D when compared to wild-type ATG4B. As discussed in the paper, we propose that S316 is involved in binding to LC3 and thus both, an alanine substitution as well as a glutamate substitution reduces the affinity to LC3. This is now included as new Figure 3b.

- 4) *Minor point. It would be helpful if the authors could indicate what band is specifically being detected by the pATG4B antibody in the cell lysate western blot (Fig 2d).*

Reply:

To better visualize the band detected by the phospho antibody in cells, we have performed an immuno-precipitation experiment with ATG4B and show the blot for phospho-S316 in Figure 2d. This blot helps to identify the band detected by the phospho-antibody, and clearly shows that there is decreased ATG4B phosphorylation on Ser316 in ULK1/2 knockout MEFs compared to wild-type counterparts. In addition, we have further characterized the antibody by treating protein preparations immuno-precipitated from cells expressing ATG4B with phosphatase (Calf intestinal phosphatase) and lysates of cells co-transfected with ULK1 that allow us to further confirm the phospho-dependency of the ATG4B signal (Supplemental Figure 2).

Reviewer #2 (Remarks to the Author):

Specific comments:

- (1) To examine the significance of ATG4B phosphorylation by ULK1 in autophagosome formation, the authors need to obtain a non-phosphorylatable mutant of ATG4B. I wonder if the replacement of Ser316 with other residues such as Thr could retain the interaction with LC3. Alternatively, mutations in residues around Ser316, a consensus motif for phosphorylation by ULK1, may impair phosphorylation without affecting the interaction with LC3.

Reply:

This is an interesting point. However, we are not able to conclusively answer this comment. Any mutation of surrounding residues is also likely to disrupt binding to LC3. Finding a mutation that impairs phosphorylation but not binding is in our opinion extremely unlikely. Also, such a mutation

(if possible at all) can impair all kinds of other functions. What we have done, however, is to demonstrate binding of LC3 to ATG4B Ser316D and demonstrated that indeed mutation of S316 reduces the affinity for LC3.

(2) *The experiments in Figure 1b is very important to show that ULK1 modulates ATG4B activity, but contain some unclear points. Why are cleaved GST and LC3 fragments observed already at time 0? Whereas the authors claim that less LC3 was generated in the presence of ULK1, the level of GST fragments appears the same between the reactions with and without ULK1. On the other hand, the authors should put a control, in which a kinase dead mutant of ULK1 is added.*

Reply:

We thank the reviewer for spotting this. In fact, this is the result of mis-labelling of the blot in the Figure. We checked against the original lab book entries and the time points used in this experiments (and the replicates) were 2, 6 and 10 min and not 0, 5, and 15 min. We have corrected this in the new Figure.

The reviewer is also right to point out that a control needed to be included, which we have done in the new Figure 1c. We have no access to a recombinant purified kinase dead mutant protein of ULK1. However, in order to address whether the observed effects are due to the catalytic activity as opposed to the mere presence of ULK1, we have assessed the LC3-GST cleavage reaction in the presence and absence of ATP. In agreement with the manuscript, we observed that LC3-GST cleavage was reduced in the presence of ATP, thus confirming that ULK1 catalytic activity is required. Further, we have included as controls a time point 0 reaction, which shows no cleavage, as well as a cleavage reaction in the presence of catalytic inactive ATG4B C74S, which also shows no cleavage. Overall, these additional experiments confirm that active ULK1 kinase activity reduces ATG4B proteolytic activity. This is shown in Figure 1c.

(3) *Does the result shown in Figure 2d mean that this antibody broadly reacts with ULK1 substrates not specifically to ATG4B phosphorylated at Ser316?*

Reply:

This is a very good observation. To confirm the specificity of the antibody, we have performed an immuno-precipitation experiment with ATG4B and show the blot for phospho-S316 in Figure 2d. Importantly this allowed us to observe a strong decrease phospho-S316 signal in ULK1/2 KO MEFs compared to WT MEF showing that ULK1 is the major kinase phosphorylating ATG4B on this site (Fig. 2d). From these experiments we also confirm, as observed by the reviewer, that non-specific bands can be observed in WT cells that show decreased signal in MEF ULK1/2 KO cells. We also provide evidence that these bands are phospho-dependent as the signal is markedly reduced after treating the lysates with phosphatase (Supplemental Figure 2). Collectively these experiments suggest that the raised polyclonal antibody can also detect non-ATG4B ULK1 dependent phosphorylated motifs, likely due to the unique motif recognized by ULK1, and we aim to make use of this property to determine whether this could be a novel tool to identify ULK1/2 target proteins. Of course, we will make this antibody widely available upon publication of the article.

(4) *In Figure 3a, the authors should quantify the bands and show the values for phosphorylated ATG4B corrected using the intensities of the Halo-ATG4B bands.*

Reply:

We have now included the normalized quantification of ATG4B phosphorylation corrected by Halo-ATG4B as suggested by the reviewer.

(5) *Whereas the author reason that loss of LC3 puncta in cells highly expressing ATG4B is due to excessive delipidation, Yoshimori's group reported that overexpression of ATG4B inhibits lipidation of LC3 (Fujita et al, 2008, MBoC).*

Reply:

We thank the reviewer for pointing this report out and have included the reference in our discussion, which, combined with our novel data showing decreased interaction between LC3 and ATG4B (Fig. 3b), has allowed us to better interpret the recovery of puncta formation that we have observed in cells highly expressing ATG4B phospho-mutants.

(6) *In the description on Figure 5b, it is unclear which protein corresponds to the regulatory 3B subunit of PP2A.*

Reply:

We are sorry for this confusion. We have marked the lanes that correspond to PP2A R3B with an asterisk. Please note that there are three different clones for PPP2R3B, all three of which show the same reduction. We have made this clearer in the Figure legend.

(7) *It would be important to test if ULK1 also phosphorylates other ATG4 paralogs.*

Reply:

Indeed, we were intrigued by the possibility whether ULK1 can also phosphorylate other ATG4 paralogues. According to the sequence alignment (Figure 2a), it can be expected that ATG4A is a potential phospho-target as well, whereas ATG4C and D are quite different in the ULK1 target motif.

To address this, we have now performed an *in vitro* kinase assay with recombinant ATG4A. We reasoned that the sequence homology of ATG4A is sufficiently similar to ATG4B and used the phospho-Ser316 antibody for assessing phosphorylation. We do not see any band detected by the pSer316 antibody, suggesting either that ATG4A is not phosphorylated on this site or alternatively suggesting that the ATG4A sequence is too dissimilar from ATG4B. Regardless, we feel that this experiment adds to a better characterization of the antibody since the reviewers raised concerns over the specificity. We can conclude that the pS316 antibody does not cross-react with ATG4A exposed to ULK1. We have included this experiment as Supplemental Figure 1. We would also like to draw

your attention to another study by the Reggiori group that has observed a similar mechanism of ATG4 phosphorylation in yeast (personal communication, submitted). However, they could not observe phosphorylation on the homology site to S316, but rather a different serine close by. Therefore, it is possible that ATG4A is indeed phosphorylated at another site and this may be a more common feature of ATG4 family of proteins.

- (8) In the text, the authors use the words autophagosome maturation probably to mean a late stage of autophagosome formation, but this would be confusing or misleading, since autophagosome maturation can be regarded to mean fusion of complete autophagosomes with endosomes/lysosomes.

Reply:

We are sorry for this confusion. We have replaced the word autophagosome maturation accordingly in the text.

Reviewer #3 (Remarks to the Author):

We thank the reviewer for the comments and agree that this study is opening up several new avenues how to think about ATG4B and LC3 processing in cells.

To make the manuscript clearer and to alleviate some of the concerns the reviewer poses, we have performed multiple additional experiments as outlined below:

1. New Figure 1c shows that ULK1 reduces ATG4B mediated processing in an ATP-dependent manner, as requested by reviewer 2.
2. New Figure 2d replaces the original Figure 2d and shows the specificity for the anti-p316 antibody in recognizing ATG4B in a co-IP experiment.
3. New Figure 3b shows a co-immunoprecipitation of ATG4B and its mutants with LC3 to show that modification of S316 reduces binding of LC3, in accordance with the model in Figure 3a.
4. New Supplemental Figure 1 shows that the p316 antibody does not recognize a phosphorylated form of ATG4A.
5. New Figure 3f shows the lipidation status of LC3 in cytoplasmic and membrane fractions with an added loading control.
6. New Figure 4c replaces the old Figure 4c and includes the S316A mutant and panels with DMSO control are shown in Supplemental Figure 3.
7. New Figure 5b replaces old Figure 5b and shows proper labeling of the cDNA constructs.
8. New Supplemental Figure 2 characterises the phospho-antibody in more detail using phosphatase treatments.

The reviewer raises the question whether phosphorylation of ATG4B by ULK1 occurs in response to a stimulus. Indeed, we were also intrigued by the possibility that phosphorylation occurs in response to a specific stimulus, and also by the possibility that phosphorylation and de-phosphorylation may be a spatio-temporally controlled event. However, this paper makes no statement about stimulus dependent or spatio-temporal control of autophagy. What we present here is evidence that ATG4B is controlled by a phosphorylation and dephosphorylation event in vitro and in cells. The paper should be seen as this and not more, since the data to date are not supporting the models that the reviewer suggests. And nowhere in the manuscript are we making such a claim.

We agree that a physiological stimulus would be nice to see, however, this is beyond the scope of this publication. The molecular regulation in terms of physiological stimuli is a new project that will not be as straight forward as one might think. For example, ULK1 controls many steps in autophagy, thus complicating any straight-forward identification of localized effects. To circumvent this, one would need to generate mutants for specific ATG proteins that “arrest” the autophagosome formation at a specific step. This has never been done in mammalian cells. Also, this requires the development of several novel tools and candidate fishing, which is not doable in any reasonable time frame. This is a problem that we aim to address in the future and something we have been carefully thinking about, but feel we are not able to provide any clear answer to soon.

Additional comments:

Figure 1A – mycULK1 KI is expressed to a lower extent compared to the wt, which might affect the activity of ATG4B.

Reply:

We recognize that ULK1 KI is slightly less expressed than wt, but this difference is marginal and does not account for the difference in ATG4B activity.

Figure 1B – LC3-GST is over exposed and it appears that its level is different between the no ULK1 and the ULK1 lanes.

Reply:

This figure does not show a blot, but it is a Coomassie gel. It was scanned with settings that are below any indication of “overexposure”. The same amounts of LC3-GST were used. ATG4B is generally so very active that it can convert all substrate very quickly. In order to see differences in the cleavage product, the reaction was set up so that enzyme (ATG4B) was used at much lower concentration than substrate (LC3-GST), as is usually the case for in vitro assays. Therefore, there is still a large amount of LC3-GST visible and differences can only be detected for the cleavage products. Furthermore, we have done the assay in the presence or absence of ATP, in order to account for differences due to the presence of ULK1 irrespective of activity and observed that in the presence of ATP (active ULK1), there is a reduction in cleavage activity, further supporting that active ULK1 reduces the in vitro

activity of ATG4B. This figure is shown as Figure 1c. This figure also includes a control that shows LC3-GST is not cleaved at time 0 (before adding ATG4B).

Figure 2C – there is lower level of ATG4B in the S316A construct compared to the wt which might affect the phosphorylation level.

Reply:

The levels of S316A vs wt ATG4B do not correlate with phosphorylation levels, and therefore do not infringe on the results seen and discussed. Furthermore, we have repeatedly seen that S316A shows no phosphorylation, and it is also shown in Figure 3c.

Figure 2D – ATG4 should be immunoprecipitated prior to testing the anti-p316 antibody to show that indeed the phosphorylated form of ATG4 is recognized.

Reply:

We thank the reviewer for this suggestion. We have performed an immuno-precipitation experiment with ATG4B and show that the phospho-S316 recognizes the phosphorylated form of ATG4. This is included in the new Figure 2d.

Figure 3C- loading control for the membranal fraction is needed.

Reply:

We thank the reviewer for pointing out that a loading control has been missing. We have now included controls for membrane and cytosol fractionation and have included this new in Figure 3e.

Figure 4C – over expression of the S316D mutant looks similar to the wt control. The differentiation to high and low expressing cells is not convincing as in the figure there are highly expressing cells with few/many spots in both wt and the mutant over expression. Furthermore, untreated control should be also presented in the figure.

Reply:

We have included the untreated control images in the panel, as suggested as a Supplemental Figure 2. We have also included the S316A mutant in new Figure 4c. The quantification of the images includes the cells that the reviewer refers to as “highly expressing with few/many spots” and the quantification

is shown in the panel. High/low expressing cells have been identified in an unbiased manner by the image analysis software using an algorithm that segments cells based on threshold. The measurements were then taken in an unbiased manner in the respective fractions. The difference that the reviewer points to is actually reflected in the quantification: wt ATG4B displays a higher value than just Halo alone, but is significantly lower than S316D. We should also note that the images displayed in Figure 4d are slightly overexposed in order to visualize both high and low expressing cells. There are still marked differences in expression level between what seems high expressing cells. During image acquisition, the Halo channel was additionally acquired at a lower laser intensity which did not over-expose the image, and this channel was used to threshold high/low expressing cells. We have the reduced intensity (non-overexposed) images as well, and have shown this in Supplemental Figure 2.

Figure 5C – Autophagic flux should be also examined in the presence of okadaic acid. The specific effect of okadaic acid on PP2A should be demonstrated and more specific genetic approaches should be taken to identify the relevant phosphatases.

Reply:

We thank the reviewer for raising awareness about the complex role of PP2A in autophagy. As the reviewer is probably fully aware, another study published in Nature Communications (Wong et al. Nat Comm 84048, 2015) has identified protein phosphatase 2A as a key regulator of ULK1. In fact, PP2A activity is required to de-phosphorylate S637, leading to activation of ULK1 and induction of autophagy. The authors of this paper have demonstrated that okadaic acid treatment leads to reduced autophagosome numbers ULK1- and ATG16 positive punctate structures and a block in LC3 turnover and GFP-LC3 puncta formation. Thus, the suggested experiment to determine autophagic flux in the presence of okadaic acid has already been published. Similarly, the authors have investigated the effect of siRNA-mediated knockdown of PP2A on autophagic flux and similarly shown that ULK1 dephosphorylation and autophagosome formation is blocked. Assessing autophagic flux in this setting will be very difficult to interpret in light of the previous findings.

Here, we show that PP2A can remove the ULK1-mediated phosphorylation of S316 on ATG4B. The referee suggests to use a genetic approach, presumably by siRNA or CRISPR. As requested, we have performed this experiment, using siRNA mediated knockdown of PP2A and assaying for phosphorylation of S316 on ATG4B. However, this assay using siRNA is complicated by the fact that PP2A knockdown will lead to inactivation of ULK1, as shown in Wong et al. As a consequence, PP2A knockdown does not result in enhanced S316 phosphorylation, but rather a reduction in S316 phosphorylation, as shown in below experiment. We have omitted this experiment from the manuscript, but again, if the reviewer feels we should add this as supplement to further clarify the role of PP2A in autophagy, we are more than happy to do so.

Overall, our view is that knockdown or knockout will not be an immediate treatment, so all effects are the cumulative effect of PP2A inhibition with regards to impaired autophagosome formation and reduced ULK1 activity. Therefore, we will not be able to attribute any observed effect to ATG4B phosphorylation status as other mechanisms are affected. The only experiment that really works here is to demonstrate that phosphorylation of ATG4B is affected by an acute inhibition of PP2A, which is what we show in the manuscript.

Reviewers' comments:

Reviewer #1 (Remarks to the Author):

The revision is suitable for publication. As a very minor point, the new Fig. 3e should be called out in the text at or near line 173.

Reviewer #2 (Remarks to the Author):

The authors have addressed some of issues I raised in the review of the original manuscript, and the manuscript has been improved to some degree. However, my two major concerns, (1) ULK1-dependent phosphorylation of endogenous ATG4B and (2) the significance of ATG4B phosphorylation in autophagosome formation, have been addressed not clearly and not at all, respectively. Without these, there remains a danger that ULK1-dependent phosphorylation of ATG4B the authors observed in this study is an artifact, and mutation analyses would merely be those of ATG4B mutants defective in the interaction with LC3 independently of phosphorylation by ULK1. Therefore, I found the revised manuscript is still too premature to be published in this journal.

Specific comments:

(1) In Figure 2d, the authors examined the phosphorylation of FLAG-tagged ATG4B (overexpressed? I could not find information about the expression level) by endogenous ULK1/2. However, immunoblotting using anti-FLAG (the bottom panel) shows that the protein level of FLAG-ATG4B in DKO cells is somehow much less than that in WT cells. This makes the interpretation of the results complicated and the conclusion unconvincing.

(2) In Supplemental Figure 1, since the protein levels of ATG4A-GST are obviously less than those of ATG4B-GST, the authors' argument about the antibody specificity is not convincing.

(3) Regarding my previous comment 8, the authors are still using "autophagosome maturation" to indicate a late stage of autophagosome formation in the introduction section.

Reviewer #3 (Remarks to the Author):

None

Reviewer #1 (Remarks to the Author):

The revision is suitable for publication. As a very minor point, the new Fig. 3e should be called out in the text at or near line 173.

Reply: We have included a reference for Figure 3e in the text now. Apologies for this oversight!
We have also re-named Figure 2d and e, as they appear in order in the text.

Reviewer #2 (Remarks to the Author):

The authors have addressed some of issues I raised in the review of the original manuscript, and the manuscript has been improved to some degree. However, my two major concerns, (1) ULK1-dependent phosphorylation of endogenous ATG4B and (2) the significance of ATG4B phosphorylation in autophagosome formation, have been addressed not clearly and not at all, respectively. Without these, there remains a danger that ULK1-dependent phosphorylation of ATG4B the authors observed in this study is an artifact, and mutation analyses would merely be those of ATG4B mutants defective in the interaction with LC3 independently of phosphorylation by ULK1. Therefore, I found the revised manuscript is still too premature to be published in this journal.

Reply: The reviewers' two main claims are 1) that we do not demonstrate ulk1-dependent phosphorylation in vivo, and 2) that the role for this phosphorylation in autophagosome formation is not clear. However, we disagree with this assessment.

Regarding point #1, there is clear evidence that addresses the ULK1-dependent phosphorylation in vivo. First, as shown in Figure 2e (previously 2d), using ULK1/2 double knockout cells, we show that the phosphorylation of ATG4B is dependent on ULK1/2 in vivo. And secondly, the phosphorylation of Ser316 in vivo has previously been shown by mass spectroscopy (PhosphoSite Plus – Imami et al. Anal Sci 24:161-6, 2008). The authors have analysed cultured cancer cells by a novel method combining phosphopeptide enrichment with nanoLC-MS/MS and identified a phosphorylated ATG4B pSer316 peptide from HeLa cells. We have included this reference and discussed this point in the text. Overall, these points taken together demonstrate that a) endogenous ATG4B is phosphorylated in vivo at position S316 by mass spectrometry; and b) that this phosphorylation is ULK1/2-dependent (by using ULK1/2 double knockout cells). We therefore conclude that we have fully addressed this point.

The second point about autophagosome formation is clearly outside the scope of our manuscript as we are merely providing a model for phosphorylation and dephosphorylation of ATG4B in cells. It is, for example, possible that the phosphorylation plays a role in non-autophagy related functions of ATG4B/LC3 and thus may have a much broader relevance beyond autophagosome formation.

Specific comments:

(1) In Figure 2d, the authors examined the phosphorylation of FLAG-tagged ATG4B (overexpressed? I could not find information about the expression level) by endogenous ULK1/2. However, immunoblotting using anti-FLAG (the bottom panel) shows that the protein level of FLAG-ATG4B in

DKO cells is somehow much less than that in WT cells. This makes the interpretation of the results complicated and the conclusion unconvincing.

Reply: As mentioned above, we show that the phosphorylation of ATG4B is dependent on ULK1/2 in vivo. The reviewer comments that due to reduced expression of ATG4B, the results are difficult to interpret. Therefore, we have added quantification which clearly shows that ATG4B phosphorylation is reduced in ULK1/2 double knockout cells when normalized to total ATG4B signal. The reduced expression is due to reduced transfection efficiency of Flag-tagged ATG4B in the double knockout cells.

(2) In Supplemental Figure 1, since the protein levels of ATG4A-GST are obviously less than those of ATG4B-GST, the authors' argument about the antibody specificity is not convincing.

Reply: We have enhanced the contrast of this gel and as is clearly evident, there is no band visible for ATG4A. However, we would like to point out that problems with interpretation of this Figure is clearly discussed in the manuscript, as the Figure does not conclusively rule out phosphorylation of human ATG4A. This is clearly stated and therefore should not be raised as a further concern.

(3) Regarding my previous comment 8, the authors are still using "autophagosome maturation" to indicate a late stage of autophagosome formation in the introduction section.

Reply: We have removed the word "maturation" throughout the manuscript.

Reviewer #3 (Remarks to the Author):

None

Reply: Thank you very much for your time and help on this manuscript.

REVIEWERS' COMMENTS:

Reviewer #4 (Remarks to the Author):

I have read both papers.

Their findings are very interesting. If their model is the case, it would impact on the field. Therefore, I agree that the authors should show ULK1-dependent phosphorylation of endogenous ATG4 in vivo in both yeast and mammals.

Response to reviewers:

Reviewer #4 (Remarks to the Author):

I have read both papers.

Their findings are very interesting. If their model is the case, it would impact on the field. Therefore, I agree that the authors should show ULK1-dependent phosphorylation of endogenous ATG4 in vivo in both yeast and mammals.

Response:

We thank the reviewer for his/her comments and appreciate that the reviewer finds the findings “interesting” that “would impact the field”. We understand the concerns regarding a demonstration of ULK1-dependent phosphorylation of **endogenous** ATG4B but the broad reactivity of the phospho antibody against total lysates in the absence of over-expressed protein has impeded this experiment. We have therefore better clarified our findings in the results and discussion clearly stating that endogenous ATG4B phosphorylation was not demonstrated.

Specifically, we included the following statement on page 11, last paragraph:

“While we do not demonstrate ULK1-dependent phosphorylation of endogenous ATG4B, we propose that cycles of phosphorylation and de-phosphorylation of ATG4B regulate the activity of ATG4B and LC3 processing.”